# A Rapid and Sensitive Aptamer-Based Biosensor for Amnesic Shellfish Toxin Domoic Acid

**DOI:** 10.3390/bioengineering9110684

**Published:** 2022-11-11

**Authors:** Luming Zhao, Han Guo, Han Chen, Bin Zou, Chengfang Yang, Xiaojuan Zhang, Yun Gao, Mingjuan Sun, Lianghua Wang

**Affiliations:** 1Department of Biochemistry and Molecular Biology, College of Basic Medical Sciences, Naval Medical University, Shanghai 200433, China; 2College of Medicine, Shaoxing University, 900th Chengnan Avenue, Shaoxing 312000, China

**Keywords:** domoic acid (DA), Capture-SELEX, aptamer, biolayer interferometry (BLI), aptasensor

## Abstract

With the incidence of harmful algal blooms (HABs) increasing in recent years, the urgent demand for the detection of domoic acid (DA), an amnesic shellfish toxin mainly produced by red tide algae *Pseudonitzschia*, has aroused increasing attention. Aptamers, a new molecular recognition element, provide clarity in the monitoring of DA. In this study, aptamers of DA were successfully screened by Capture-SELEX. Through identification and truncation optimization, aptamer C1-d with a high affinity (*K*_D_ value, 109 nM) and high specificity for DA was obtained. The binding mechanism between DA and the aptamer was explored by molecular docking and molecular dynamics (MD) simulation, revealing the critical sites for DA–aptamer interaction. Meanwhile, a BLI-based aptasensor was constructed by C1-d, which displayed a linear range from 0.625 to 10 μM and a LOD of 13.7 nM. This aptasensor exhibited high specificity, good precision and repeatability, and high recovery rates for real samples; the process of detection could be completed in 7 min. This study is the first to identify and investigate the binding mechanism of DA–aptamer interaction and constructed a BLI-based aptasensor for DA, which lays a theoretical foundation for the detection and prevention of DA.

## 1. Introduction

In 1987, a serious food poisoning outbreak occurred in Canada. More than 100 residents contracted food poisoning through eating contaminated seafood. Victims showed gastrointestinal symptoms (nausea, vomiting, and diarrhea) and chronic nerve damage (persistent memory impairment, disorientation), and some cases even led to coma and death [1,2]. Since then, the term amnesic shellfish poisoning (ASP) has been used clinically. Experts found that the substance responsible for this poisoning was domoic acid (DA) and named it amnesic shellfish toxin (AST). As harmful algal blooms (HABs) have occurred throughout the world, DA has been found to have a global distribution in many countries such as North and South America, Australia, Europe, and Asia [3,4].

Domoic acid (C_15_H_7_NO_6_, molecular weight 311.33) is a kind of small-molecular excitatory proline derivative and potent neurotoxin, which is mainly produced by the *Pseudonitzschia* genus. After being ingested by aquatic organisms, DA is enriched through the food chain, thus threatening human health. Importantly, the heat stability of DA makes its toxicity difficult to overcome through ordinary cooking methods. In addition, there is no effective treatment for the poisoning induced by DA. Domoic acid contains three carboxyl groups and a secondary amino group, and its form in solution is easily affected by pH. There are eight isomers of DA (A–H). These isomers may be the reaction products of DA exposed to ultraviolet light, rather than the natural products of algae. The main contaminant of marine and aquatic organisms is DA, which is the most toxic of all these isomers [5,6,7]. There is a great similarity between the structures of DA, glutamic acid (Glu), and kainic acid (KA). However, the excitatory effect of DA on nerve cells is more than 100 times higher than Glu and 2–3 times higher than KA [8,9]. Domoic acid is a stimulant that can directly or indirectly activate glutamate receptors, including the N-methyl-D-aspartate (NMDA) receptor, α-amino-3-hydroxy-5-methylisoxazole-4-propionate (AMPA) receptor, and kainate or kainic acid receptor. These receptors distribute in different regions of the central nervous system and are activated by DA under different physiological conditions. Domoic acid can cause damage to neurons in the hippocampus and neocortical neurons [8,10,11]. It has been found that when the content of DA in shellfish reaches 40 mg/kg, it can cause poisoning, and 150 mg/kg leads to a risk of death. The acute reference dose (ARfD) of an adult is 30 μg DA/kg b.w. The European Union stipulated that the DA content should not exceed 20 mg/kg in shellfish, and the European Food Safety Authority (EFSA) recommended that the safe edible limit of DA be 4.5 mg/kg in shellfish [12]. High toxicity, quick onset of symptoms, and the lack of suitable antidote bring great challenges to the detection and prevention of DA.

Mouse bioassay (MBA) for DA was developed based on the AOAC official method 959.08 of paralytic shellfish poisoning (PSP), in which toxicity was evaluated according to the survival time and poisoning symptoms of mice after injection, which was shown as a half-lethal dose. Despite being the earliest investigation of DA poisoning in 1987, this method was deemed unsuitable for DA monitoring because of its high detection limit, high variability, low sensitivity, and the necessity of using mouse species [12]. High-performance liquid chromatography (HPLC) is the most widely used method for DA detection and has been listed as a national standard method in many countries due to its high sensitivity and high accuracy. High-performance liquid chromatography–ultraviolet detection (HPLC-UV) is particularly suitable for the determination of DA content in shellfish tissues, especially when the toxin content exceeds 20 μg/g. Quilliam et al. reported a rapid extraction and cleanup for the liquid chromatographic determination of DA in unsalted seafood [13], which was used as a reference for HPLC-UV methods around the world. Lawrence et al. improved the HPLC-UV method by a reverse-phase C18 column to detect DA at 242 nm [14]. López-Rivera et al. established an HPLC-UV method for DA detection without SPE pretreatment, in which compounds were separated by equal gradient leaching and pH control of the mobile phase [15]. However, the low pH is not conducive to the stability of DA, and neither is the MBA method. Although these methods are sensitive and stable, and have low detection limits, they require cumbersome equipment, extensive use of chemicals, and highly trained personnel to carry out the complex and time-consuming process. Enzyme-linked immunoassay (ELISA) is a biochemical assay established based on the highly selective reaction between antibody and antigen or hapten [16], which has the advantages of sensitivity, convenience, and batch detection. However, it is mainly used for preliminary screening of DA due to cross-reaction. Additionally, the high cost of standards and small molecular weight of DA make it more difficult to prepare immune antigens, which limits the application of immunological methods in DA analysis [17,18,19,20]. Capillary electrophoresis (CE) is a liquid-phase separation and analysis technique based on various characteristics of the sample (charge size, isoelectric point, polarity, etc.), and then detected by fluorescence or UV detector. Zhao established a capillary electrophoresis-ultraviolet detection method to detect DA in seafood [7]. The online capillary isokinetic electrophoresis–capillary zone electrophoresis method was established to detect DA in shellfish and algae [21]. Zhang quantified DA in shellfish samples by capillary electrophoresis-based enzyme immunoassay with electrochemical detection [22]. CE methods can be automated and enable high-resolution separation of DA and its isomers. However, they have the disadvantage of requiring both cleanup steps to achieve reliable results and highly trained personnel. In brief, there are still some limitations of the current DA detection techniques. Therefore, it is necessary to establish efficient methods and explore new molecular recognition elements of DA in order to safeguard the marine environment, seafood quality, and human health.

The term aptamer refers to a molecule with a specific tertiary structure and certain functions formed by self-crimped folding of single-stranded DNA or RNA in the absence of complementary strands. Aptamers are generally obtained by the systematic evolution of ligands by exponential enrichment (SELEX) [23,24]. Aptamers and their targets form complexes with relatively stable spatial structures mainly through hydrogen bonding, hydrophobic interaction, electrostatic interaction, and van der Waals force [25,26]. Aptamers can specifically recognize a wide range of targets including small molecules, proteins, pathogens, viruses, cells, tissues, and even living animals [27,28,29,30,31,32,33,34,35]. Aptamers have similar or even better performance than antibodies. In addition to high affinity and high specificity, they also possess many characteristics such as low cost, chemical and thermal stability, low immunogenicity, easy chemical modification, and easy synthesis in large quantities [36,37,38]. As novel molecular recognition elements, aptamers have prospective applications that cover environmental monitoring, food safety, diagnosis, drug delivery, and treatment [29,39,40,41,42,43]. Currently, specific aptamers towards different kinds of marine biotoxins have been obtained, such as saxitoxin (STX), okadaic acid (OA), tetrodotoxin (TTX), brevetoxin-2 (BTX-2), and nodularin-R (NOD-R) [44,45,46,47,48]. Therefore, it is possible to screen out aptamers and apply them in the detection and prevention of DA.

## 2. Materials and Methods

### 2.1. Materials and Reagents

All ssDNA oligonucleotides were synthesized and purified by high-performance liquid chromatography (HPLC) by Sangon Biotechnology Co., Ltd. (Shanghai, China). DA and KA were purchased from the National Research Council Canada (Halifax, NS, Canada). OA, STX, TTX, NOD-R, dinophysistoxin (DTX), and gonyautoxin (GTX) were purchased from Taiwan Algal Science Inc. (Taiwan, China). Dynabeads^TM^ MyOne^TM^ Streptavidin C1 (diameter 1.0 μm) and the Qubit^®^ ssDNA assay kits were purchased from Thermo Fisher Scientific (Chelmsford, MA, USA). GoTaqHot^®^ Start Colorless Master Mix was purchased from Promega Corporation (Fitchburg, WI, USA). QIAEX^®^ II gel extraction kits were purchased from Qiagen (Frankfurt, Germany). Super streptavidin (SSA) sensor biosensors were purchased from Sartorius (Shanghai, China). Selection Buffer (pH 7.4, 8 g/L NaCl, 2.8975 g/L Na_2_HPO_4_, 0.1 g/L MgCl_2_·6H_2_O, 0.2 g/L KH_2_PO_4_, 0.2 g/L KCl, and 0.1 g/L CaCl_2_) was purchased from Tiandz (Beijing, China). All reagents were of analytical grade and were used without further purification or treatment unless otherwise specified. All the solutions were prepared with ultrapure water.

### 2.2. Instruments

The amplification was carried out using a PCR instrument (Bioer Technology, Hangzhou, Zhejiang, China). The incubation was conducted in tubes using a four-dimensional rotary instrument (Kylin-Bell Lab Instruments Co., Ltd., Nantong, Jiangsu, China). The DNA concentration was quantified using a Qubit^®^ 3.0 fluorometer (Life Technologies, Carlsbad, CA, USA). The Milli-Q water purification system was purchased from Millipore Corp (Bedford, MA, USA). BLI assay was performed using an OctetRED K2 instrument (Sartorius, Shanghai, China).

### 2.3. Aptamer Selection by Capture-SELEX

#### 2.3.1. The ssDNA Library and Primers 

All the DNA sequences in this research are shown in Appendix A. The ssDNA library of Capture-SELEX was 80 nt in length, which included two constant regions of 20 nt located, respectively, at the 3′ end and 5′ end, and a 40 nt random sequence lies between the constant regions. Constant regions were used for primer binding to amplify the next round of library. The polyadenine-modified primer R1 promoted the separation of PCR duplexes into single strands of different sizes. Biotinylated capture oligo can be bonded to the ssDNA library by base complementary pairing. Due to the interaction between biotin and streptavidin, the ssDNA could be immobilized to streptavidin beads by capture oligo. 

#### 2.3.2. Hybridization and Immobilization of the ssDNA Library 

At the beginning of each round, the ssDNA library and biotin-modified oligonucleotide should be hybridized in a ratio of 1:2. The hybridization requires the following steps: 95 °C for 10 min, 60 °C for 60 s, then cool slowly to 25 °C for 60 s at a rate of 0.1 °C/s. The hybrid of the ssDNA library and biotin-modified oligonucleotide was incubated with streptavidin magnetic beads for immobilization. The DNA concentration of the hybrid was quantified by a Qubit^®^ 3.0 fluorometer and marked as C1. The hybrid was mixed with magnetic beads and incubated on the four-dimensional rotary instrument. After 1 h of incubation, all magnetic beads were absorbed with magnets, and DNA concentration in the supernatant was measured, denoted as C2. The value of C2/C1 was used to calculate the immobilization efficiency of the ssDNA library. If C2/C1 is less than 0.5, the immobilization is not sufficient. In this study, the library immobilization efficiency was above 85% in each selection round.

#### 2.3.3. Screening Procedure 

The magnetic-bead-immobilized ssDNA library was washed at least five times by selection buffer to remove unbound or infirmly bounded sequences. Subsequently, the target toxin was added to the screening system and incubated with magnetic beads at room temperature. Sequences that bound specifically to the target were eluted from the beads. The concentration of the ssDNA elution was measured to calculate the recovery rate of each round, which is the ratio of the amount of ssDNA obtained from the elution to the input amount of the ssDNA library. To prepare the secondary library, the eluted DNA was amplified by PCR and the short-strand ssDNA of the PCR duplex was purified by gel extraction.

The selection process and its details are illustrated in Figure 1 and Appendix A. To improve the efficiency and specificity, the screening pressure was increased by the addition of counter target KA, shortening the incubation time of DA, and prolonging the incubation time of KA from round 6 to round 12.

### 2.4. Sequencing of ssDNA

The ssDNA products eluted by the target from rounds 1, 3, 4, 6, 8, 10, and 12 were amplified by forward primers with different tags (T1–T7) and the reverse primer R2, respectively (Appendix A). PCR products from different rounds were mixed in equal quantities, purified, and sent for high-throughput sequencing (HST). The ssDNA elution in the 12th round was amplified by the primers F1 and R2 for cloning sequencing. Sequences from HST and cloning sequencing were analyzed by Clustal X 2.1 for multiple sequence alignment and by mfold for secondary structure prediction and Gibbs free energy prediction.

### 2.5. Interaction Mechanism of Aptamer and DA

The DNA sequence of the aptamer was submitted to the RNAfold web server [49]. The 3D structure of the corresponding RNA was obtained by submitting the secondary structure and sequence on RNA Denovo [50]. Finally, the corresponding RNA 3D structure was converted into the DNA 3D structure of the aptamer by AMBER16 [51]. 

Molecular docking of DA and the aptamer was performed by the molecular operating environment (MOE) dock. Diagrams of intermolecular binding patterns were completed by PyMOL. The MD simulations were performed by AMBER16 software. Specific parameter settings and software are shown in the Appendix A.

### 2.6. Biolayer Interferometry (BLI) Assay

#### 2.6.1. Preparation of Sensors and Aptamers

The SSA sensors were immersed in the selection buffer for at least 10 min before detection. Aptamers were dissolved in selection buffer and diluted to 1 μM, and then were denatured at 95 °C for 10 min and left on ice for 5 min. 

#### 2.6.2. Detection Procedure of BLI

There were five key steps in the BLI assay in this research: baseline for 1 min; aptamer loading for 3 min; baseline-2 for 1 min; association of the aptamer and DA for 1 min; dissociation of the aptamer and DA for 1 min. The entire operation was carried out in a 96-well plate at a room temperature of 25 °C. The steps of the baseline and baseline-2 operated in the selection buffer were used for balancing. The aptamer loading step was a procedure of biotin-modified aptamers immobilizing on the SSA sensors. After the loading step, baseline-2 was prepared for the subsequent association step, which contained targets dissolved in the selection buffer. The dissociation step that operated in the selection buffer was a procedure of separation of aptamers and the target. 

### 2.7. Property Assessment of Aptasensor

The calibration curve describing the relationship between response value and concentrations was fitted to the sigmoidal logistic four-parameter equation: y = (R_max_ − R_min_)/ [(1 + (x/EC_50_) ^b^)] + R_min_

In this equation, R_max_ and R_min_ represent the maximum and minimum of the response value, respectively. EC_50_ is the concentration of DA that corresponds to half of the maximum response value and b stands for the Hill slope of the curve. The limit of detection (LOD) and the limit of quantification (LOQ) were obtained by detecting 20 blank samples (Appendix A) and were calculated by the following equations:LOD = 3 S_a_/b; LOQ = 10 S_a_/b

Here, S_a_ means the standard deviation of the response of the blank samples (*n* = 20) and b stands for the slope of the calibration curve. Precision was assessed by the coefficient variation (CV) of different concentrations of DA (*n* = 3). The reproducibility assessment was performed by detecting the 6.25 μM DA samples 10 times, and the CV values were calculated.

### 2.8. Treatment of Real Samples

#### 2.8.1. Seawater Samples

Seawater samples (2 mL) spiked with different concentrations of DA were centrifuged at 6000× *g* rpm for 10 min. The supernatants were evaporated by the concentrator, and then resuspended with 500 μL selection buffer, filtered by 0.45 μm filters.

#### 2.8.2. Shellfish Samples

DA was added to 5 g shellfish homogenate in the 50 mL plastic centrifuge tube at room temperature for 1 h. A total of 20 mL of methanol/water (1:1 *v*/*v*) was added to the mixture and then blended using a vortex mixer for 3 min, before being centrifuge for 10 min at 3600× *g* rpm. The supernatant was evaporated and resuspended with 500 μL selection buffer, and then filtered by 0.45 μm filters and stored at 4 °C.

## 3. Results and Discussion

### 3.1. Selection of DA Aptamer by Capture-SELEX In Vitro

Although DA is an important marine biotoxin, there are few reports on its specific high-affinity aptamer. It is possible that some of its properties make it challenging to screen aptamers. Firstly, DA is a kind of amino acid derivative with a molecular weight of only 311.33, and although it contains three carboxyl groups and a secondary amino group, it is inapposite to screen its aptamers by modifying and immobilizing DA onto a solid medium. The natural structure and physicochemical properties will be directly changed if DA is immobilized by magnetic beads coupled with amino groups. In contrast, screening aptamers by immobilizing the ssDNA library and allowing DA in the free state could increase the success rate of selection. Consequently, Capture-SELEX was used for aptamer screening in this research. 

Biotinylated capture oligo and ssDNA library were paired in the selection buffer through annealing. Since biotin can interact with streptavidin (SA), the ssDNA was immobilized to SA magnetic beads by capture oligo after incubation, as shown in Appendix A. In this study, the ssDNA library was designed as sequences with constant primer regions at both ends that could partially complement each other to form a stem-loop structure, which aimed to make ssDNA form a stable closed-loop structure. The complementarity of primer regions is conducive to the full exposure of random sequence for target combinations. The primer hardly participates in folding, which facilitates the subsequent truncation optimization of aptamers. The key steps of Capture-SELEX in this study are shown in Figure 1.

During the selection process, the amount of ssDNA library items, the amount of target, incubation time, and wash times after incubation should be strictly controlled (Appendix A). To improve screening specificity, the counter selection was operated by adding a counter-target KA with a similar chemical structure to DA in rounds 6–12. Counter selection facilitated the removal of non-specific sequences of DA. To improve the screening efficiency, the selection pressure was gradually increased by reducing the incubation time of positive target DA, reducing the input of the ssDNA library, and increasing the input and incubation time of counter-target KA (Figure 2A). After 12 rounds of screening, the ssDNA recovery ratio of positive selection was no longer significantly increased (Figure 2B), which was regarded as reaching the screening endpoint. Fortunately, the recovery ratio of KA gradually decreased, which proved the effectiveness of counter selection. The ssDNA eluted in the positive selection was amplified by PCR, followed by cloning sequencing, high-throughput sequencing, and further analysis.

### 3.2. Acquisition and Affinity Identification of Candidate Aptamers

Eighty clones were randomly selected from the 12th round of ssDNA sequences for cloning sequencing. Names of sequences were selected by the number of randomly selected clones. In the results of clone sequencing, C1 was found 13 times, C100 repeated 12 times, C12 repeated 5 times, and C58 repeated 2 times, which could be attributed to sequence enrichment as the selection round went on. Meanwhile, we also performed HTS, which could track the enrichment trajectory of aptamers throughout the whole screening process. As shown in Appendix A, we analyzed the abundance enhancement of the top 10, 100, and 1000 sequences in the results of HTS. The sequences were enriched gradually with each round. After 12 rounds of selection, the enrichment degree of the top 10, top 100, and top 1000 sequences in the total sequence of HTS results were 83.2%, 92.7%, and 98.8%, respectively. In fact, the enrichment degree of the top 10 and top 100 reached the highest (89.5% and 96.2%) level in the eighth round. 

We named the sequences htC1, htC2, htC3, and so on, according to the order from high to low of the total number of enriched reads in HTS. Not surprisingly, the sequence htC1 with the highest total enrichment (422,409 reads) was identical to C1 obtained by clonal sequencing. Additionally, htC2 (101,246 reads), htC3 (93,846 reads), and htC4 (43,938 reads) were consistent with C12, C100, and C58, respectively, obtained by cloning sequencing. However, it is not necessarily the case that the more enriched the aptamer, the stronger its affinity for the target; thus, the level of affinity should be further verified. 

The top 100 sequences from high-throughput sequencing and 80 sequences from cloning sequencing were analyzed by Clustal X 2.1 (Appendix A) and divided into several families. One or two representative sequences with high enrichment or low Gibbs free energy from each family were selected as candidate sequences for affinity determination (Table 1). BLI results showed that among all the 10 candidate aptamers, 8 sequences had an affinity for DA, and 2 sequences showed no binding to DA. C1 was the aptamer with the highest affinity (*K*_D_ value, 1.69 × 10^−6^ M). In addition, a random sequence, which has no affinity for DA, was used as an aptamer control.

### 3.3. Truncation of Aptamer C1

To remove non-essential nucleotides, obtain the core sequence, and explore the binding mechanism, truncation optimization of the aptamer is indeed necessary [52]. It is generally believed that the truncation optimization of the aptamer can not only improve its affinity but also reduce the cost of synthesis. The ssDNA library in this study could form a closed stem-loop structure [53]; the self-complementation of primer sequences facilitated the fully exposed random sequence to combine with the target. Since primer regions did not participate in folding, they made little contribution to target binding. Therefore, the primer regions of aptamer C1 were directly removed to generate C1-s. 

BLI results showed that the affinity of C1-s (*K*_D_ value, 1.50 × 10^−6^ M) and C1 for DA was at an equivalent level. It was further confirmed that the primer regions were not core sequences for the binding of DA and the aptamer. Subsequently, we analyzed the secondary structure of C1-s using mfold web server software (http://www.unafold.org/mfold/applications/dna-folding-form.php (accessed on 8 October 2022)) (Figure 3), and the two stem-loops in C1-s were separately truncated or retained to obtain the variants aptamers, C1-a, C1-b, C1-c, and C1-d, respectively (Appendix A). All of these aptamers displayed considerable affinity with *K*_D_ values ranging from 1.09 × 10^−7^ to 6.90 × 10^−6^ M (Appendix A). The affinity between C1-d and DA was significantly increased by truncation, whereas the affinities of C1-a and C1-c were lower than C1-s. This could be due to the disruption of the DA binding sequence here. Based on the secondary structures of C1-a, C1-b, C1-c, and C1-d, we speculated that the corresponding stem-loop in C1-d was more necessary for the interaction between DA and the aptamer. In brief, C1-d was the best aptamer in truncated candidates in this study, with the highest binding affinity. Through truncation optimization, the *K*_D_ value changed from 1.69 × 10^−6^ to 1.09 × 10^−7^ M.

### 3.4. Identification of Affinity and Specificity of Aptamer C1-d

Different representative species of marine biotoxins were used to detect the affinity and specificity of C1-d to DA (Figure 4). GTX, STX, and TTX belong to alkaloids. NOD-R belongs to peptide toxin. DTX and OA are polyether toxins. KA is an analog of DA because of structural similarity. In addition, a random sequence was used for a negative control of the aptamer. Biolayer interferometry assay illustrated that C1-d possessed a *K*_on_ (1/Ms) value of 2.94 × 10^5^, a *K*_dis_ (1/s) value of 5.13 × 10^−2^, and a *K*_D_ (M) value of 1.09 × 10^−7^ M in the interaction with DA. Unsurprisingly, GTX, STX, TTX, NOD-R, DTX, OA, and KA had almost no response in the detection. In addition, there was no interaction response between the random sequence and DA. These results revealed that aptamer C1-d could bind to DA with high affinity and high specificity. 

### 3.5. Molecular Docking and Molecular Dynamics Simulations 

Although the secondary structure predicted by the mfold software helped to complete the truncation optimization of the aptamer, and the affinity test verified the best aptamer C1-d of the four truncated candidates, the binding mechanism between DA and the aptamer has not yet been clarified. To further explore the interaction between DA and the aptamer and to verify the reliability of the truncation optimization, that is, if the key sites of interaction are missed in C1-d, we will adjust the optimization strategy to obtain aptamer with higher affinity. Therefore, with aptamer C1-s, molecular docking and MD simulations were performed to resolve the abovementioned confusion.

Firstly, the 3D structure model of aptamer C1-s was obtained by RNA Denovo; the drawn view and surface view are shown in Figure 5A,B. MOE-Dock simulation was performed to determine the interaction of DA and C1-s. The docking score was −7.3191 Kcal/mol. In fact, the more negative the docking score, the stronger the binding affinity between DA and the aptamer. 

To find a stable structure of DA–aptamer, we carried out all-atom, explicit water (MD) simulations. Monitoring the root mean square deviation (RMSD) of the initial simulated structure is a common method to analyze the structural stability of biological macromolecules in the process of dynamic simulation. As shown in Figure 5C, the RMSD of the backbone of the aptamer was less than 15.0 angstrom and the DA was less than 2.5 angstrom. The system achieved equilibrium within the simulation time, which suggested the acceptability of the force field and simulation protocol. Figure 5D shows the root means square fluctuation (RMSF) value of the DA–aptamer. RMSF can indicate the changing amplitude of each atom relative to its average position, characterizing the flexibility and movement intensity of DNA bases in the simulation. In the simulation process, the RMSF value of 10–15 bases in C1-s was higher than other bases, indicating their high flexibility and instability.

MD trajectory adopted a clustering strategy. The cluster center after the equilibrium of the system was selected as the final stable structure of DA–aptamer, which is shown in Figure 5E,F. The C_1_, G_26_, and C_34_ in the aptamer formed hydrogen bonds with DA. As shown in Appendix A, the conformational changes were not obvious in the DA–aptamer, indicating that the complex did not separate during MD simulation. Additionally, the complex formed by the aptamer and DA was stable. In addition, the radius of gyration and mass distance between the aptamer and DA changed very little throughout the simulation time (Appendix A), which also indicated that the binding of the aptamer and DA was highly stable. 

The binding energy (ΔGtotal) of the DA–aptamer was calculated by MM-PBSA (Appendix A). Contributions to the binding free energy (ΔGtotal) from the VdW and electrostatic interactions were represented by ΔEvdw and ΔEele. The polar and nonpolar solvation energy contributions to ΔGtotal were denoted by ΔGpolar and ΔGnonpolar, respectively. ΔGsol is the total of polar and nonpolar solvation energy. The DA–aptamer binding was mainly due to the contribution of electrostatic interactions. ΔEelec was the most favorable for binding. ΔGpolar was an unfavorable contributor, while ΔGnonpolar was favorable for binding, which produced an overall favorable binding energy. The binding free energy of DA and C1-d was calculated to be −13.5053 kcal/mol in aqueous environments. Further energy composition analysis suggested that C_1_, G_26_, G_29_, T_30_, A_33_, C_34_, and C_36_ in aptamer C1-s contributed the most to the complex binding free energy (Appendix A). 

All the critical bases of DA–aptamer interaction were present in C1-d, while G_26_, G_29_, T_30_, A_33_, C_34_, and C_36_ appeared in the sequence of C1-b, and C_1_ and G_26_ were in the sequence of C1-c, but none of the abovementioned bases appeared in C1-a (Appendix A). The simulation results explained the reason why C1-d showed the highest affinity among the four truncated aptamers; C1-b was the second, C1-c was the third, and C1-a had the lowest affinity (Appendix A). Despite the absence of key bases in C1-a, it could bind to DA with a relatively low affinity. It was speculated that C1-d which contains the key bases could bind to DA stably, while C1-a showed an unstable interaction with DA. In the simulation process, the RMSF values (Figure 5D) of the bases 10–15 of C1-s (located in C1-a) were relatively high, indicating that they had more flexibility and presented in an unstable state. The binding of C1-a to DA might depend on the role of the less flexible bases 2–9 and bases 16–24 in C1-s. Furthermore, the key bases including G_26_, G_29_, T_30_, A_33_, C_34_, and C_36_ except for C_1_, all appeared in the stem-loop of C1-d. However, when the truncation optimization only retained only this stem-loop, namely C1-b, its affinity for DA was not as high as C1-d. On the one hand, the first base C_1_ at the 5′ end of C1-d made a huge contribution to the interaction of the DA–aptamer; on the other hand, bases at the 3′ end of C1-d could protect the critical sites in the stem-loop structure, thus making the binding more stable. Therefore, without further truncation, C1-d was used for the subsequent aptasensor preparation.

### 3.6. BLI-Based Aptasensor for DA Detection

BLI is a non-labeling technique established on the principle of optical interference. It has the advantages of easy operation, rapid and real-time detection, and less sample consumption [54]. Through the real-time monitoring of optical interference signals, the BLI technique has been widely applied in the rapid detection of biomolecular interaction analysis [55,56,57,58,59]. The white light emitted from the spectrometer is reflected at the biofilm layer interface of the biosensor, and the reflected light generated by the two interfaces forms interference light. When the target binds or dissociates on the biosensor, the thickness and mass density of the biofilm layer will change, and the interference mode of interference light will also change accordingly. The relative displacement value (Δλ) of the interference spectrum will change with time [54]. The working principle of BLI is shown in Figure 6.

We combined the specific high-affinity aptamer C1-d with biolayer interferometry technology to establish a label-free and real-time aptasensor for DA detection. When C1-d was immobilized on an SSA sensor, the target–aptamer interaction promoted DA to bind to the sensor surface, resulting in a shift of the interference spectrum. This phenomenon produced a curve of the response value.

To assess the performance of this aptasensor, the real-time responses of DA in the concentration range of 0.625–30 μM were observed. As the concentration of DA increased, the density and thickness of the biofilm surface changed greatly, and the response value increased significantly (Figure 7A). The measured data were fitted as the following equation: y = (0.4873 − 0.01530)/[(1 + (x/12.57) ^−1.351^)] + 0.01530 (R^2^ = 0.9951) (Figure 7B). Moreover, the curve showed a linear detection range of 0.625 to 10 μM DA, which can be fitted by a linear regression equation: y =0.02112x − 0.01192 (R^2^ = 0.9912) (Figure 7C). After calculation, we found that the LOD of this aptasensor was 13.7 nM, and the LOQ was 45.7 nM. As shown in Appendix A, the aptasensor was experimentally able to detect DA concentrations around (or below) the theoretical LOQ. Moreover, the aptasensor was used to detect different concentrations of DA, and the CV values were 1.12%–4.71% (Table 2). The reproducibility of this aptasensor was identified by detecting responses of 6.25 μM DA 10 times. Additionally, the average of the 10 response values was 0.1469 nm (CV, 4.06%), confirming the good reproducibility of the aptasensor (Appendix A). There were eight marine toxins (DA, GTX, STX, TTX, NOD-R, DTX, OA, and KA) used to identify the specificity of the aptasensor. A mixture of these toxins was also tested (Figure 7D). Both the DA sample and the mixture containing DA showed significantly higher responses than those seven other toxins. These results indicated that the aptasensor in this study had high precision, good reproducibility, and high specificity in DA detection. It can be used sensitively even in a complex environment containing interference from other toxins.

To evaluate the practicality of aptasensors in real detection, seawater and shellfish samples spiked when different concentrations of DA were prepared. As shown in Table 3, recovery rates ranging from 95% to 110.3% were obtained, and the CV values were within normal limits. The standard curves of seawater and shellfish detection are shown in Appendix A. These results indicate that real samples had no significant interference in DA detection. Therefore, this BLI-based aptasensor has great potential to be used in the detection of real samples contaminated by DA. Based on the seawater treatment method adopted in this study, the LOD, LOQ, and linear range of this aptasensor in seawater detection can be calculated as 1.066 ng/mL, 3.557 ng/mL, and 48.6–778.3 ng/mL, respectively. In addition, the LOD, LOQ, and linear range of this aptasensor in shellfish detection can be calculated as 0.427 μg/kg, 1.423 μg/kg, and 19.5–311.3 μg/kg, respectively, according to the shellfish treatment method in this research. 

The HPLC methods for DA detection have some limitations, including complicated and time-consuming operation, extensive use of chemical reagents, and difficulty in the simultaneous detection of multiple samples. ELISA assays sometimes cannot avoid cross-reaction. Additionally, the preparation of DA immune antigen is difficult and costly. CE requires cleanup steps to achieve reliable results and highly trained experimental skills. In addition to the above traditional DA detection methods, a few biosensors of DA have also been reported. An immuno-based screening method was developed to detect DA in extracts of shellfish species using a surface plasmon resonance-based optical biosensor. Its detection limits are in the μg/g range and performed most accurately at the European Union’s official action limit for DA of 20 μg/g [60]. Stevens developed a portable SPR biosensor for the detection of DA in shellfish, algae, and seawater. It has a limit of detection of 3 ppb and a quantifiable range from 4 to 60 ppb [61]. Lotierzo synthesized a molecularly imprinted polymer (MIP) film for DA by direct photo-grafting onto a gold chip, which is suitable for a surface plasmon resonance (SPR)-based bioanalytical instrument system [62]. However, these detection methods have not been verified in inter-laboratory studies and have not been used in wide application.

Although the LOD (1.066 ng/mL) of this aptasensor in seawater sample detection is higher than some detection methods (Table 4), the LOD (0.427 μg/kg) of shellfish detection is significantly lower than the DA content (20 mg/kg of shellfish) stipulated by the European Union and the safe edible limit (4.5 mg/kg of shellfish) recommended by the European food safety authority [12]. Therefore, this aptasensor can still be applied to DA detection. More importantly, this BLI-based aptasensor offers several breakthroughs compared with existing detection methods. Firstly, there are no complex experimental techniques, and the whole detection process can be finished within 7 min, which is rapid and convenient. Moreover, the multiple channels and 96-well plates of the BLI platform allow efficient detection for multiple samples to be examined simultaneously. Thirdly, the low cost of aptamers and reusability of aptasensors can decrease spending on experimental consumables. Fourthly, BLI can achieve automatic real-time detection by setting up an instrument program. Lastly, there is no need for microfluidics due to the direct interaction between the BLI sensor and the sample, which contributes to the integrity of the sample for easy reuse. To sum up, the BLI-based aptasensor possesses some superior qualities, such as being rapid, simple, efficient, low-cost, and high-throughput, and it is expected to be a powerful detection method of DA.

## 4. Conclusions

In this research, DA aptamers were successfully selected by Capture-SELEX. Through truncation optimization, an aptamer named C1-d with a *K*_D_ value of 109 nM was finally obtained. Additionally, it showed high affinity and high specificity for DA, as identified by the BLI assay. The binding mechanism of DA to the aptamer and their key sites of interaction were predicted by molecular docking and MD simulation, which further confirmed the high affinity and high specificity between C1-d and DA. Furthermore, we constructed a BLI-based aptasensor by C1-d, which has high specificity, good precision, good repeatability, and an acceptable recovery rate for DA detection. In brief, a rapid and sensitive BLI-based aptasensor was established in this work, which provides a new horizon for efficient monitoring of DA and lays a foundation for the application of aptasensors in the detection of marine biotoxins.

Domoic acid is widely distributed almost all over the world. It has become a serious threat to the marine environment and human health because of its high toxicity, fast-acting nature, and lack of a suitable antidote. Aptamers, as a new molecular recognition component, have great potential for the detection and prevention of marine biotoxins worldwide. They have the potential to establish rapid, specific, efficient, and sensitive detection technologies and to provide a clinical basis for the research of targeted drugs and the treatment of marine biological toxins. In future research, the aptamers used in this research will be attempted to be made into other types of kits and biosensors for DA detection. They are also expected to be used in the DA separation and enrichment reagents for eliminating disease-causing biohazards or efficiently adsorbing harmful pollutants in water.

## Figures and Tables

**Figure 1 bioengineering-09-00684-f001:**
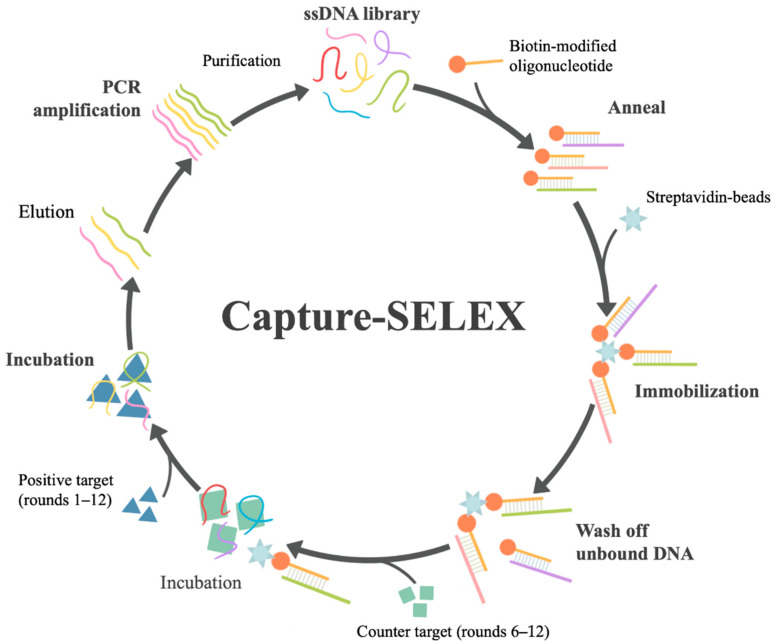
Key steps of Capture-SELEX. Each round of Capture-SELEX in this study mainly includes the following steps: (1) preparation of ssDNA library; (2) hybridization of ssDNA library and biotin-modified capture oligonucleotide; (3) immobilization of the hybrid on streptavidin beads; (4) washing off unbound ssDNA library; (5) incubation with the target (positive target domoic acid (DA) was used in rounds 1–12, negative target kainic acid (KA) in rounds 6–12) and obtaining ssDNA elution; (6) PCR amplification of ssDNA elution and purification to obtain the next-round ssDNA library.

**Figure 2 bioengineering-09-00684-f002:**
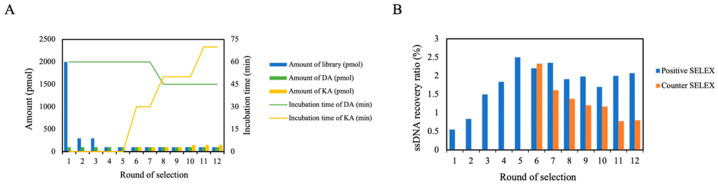
(**A**) Selection conditions of Capture-SELEX. (**B**) The recovery ratio of ssDNA in Capture-SELEX. The recovery ratio represents the percentage of the amount of ssDNA eluted by the target to the amount of library immobilized in this round.

**Figure 3 bioengineering-09-00684-f003:**
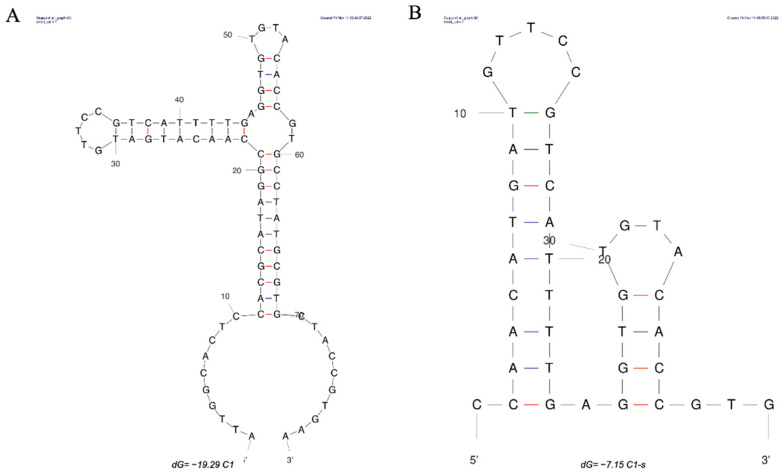
Secondary structure prediction of aptamer C1 (**A**) and aptamer C1-s (**B**) with their respective lowest Gibbs free energy value by mfold software.

**Figure 4 bioengineering-09-00684-f004:**
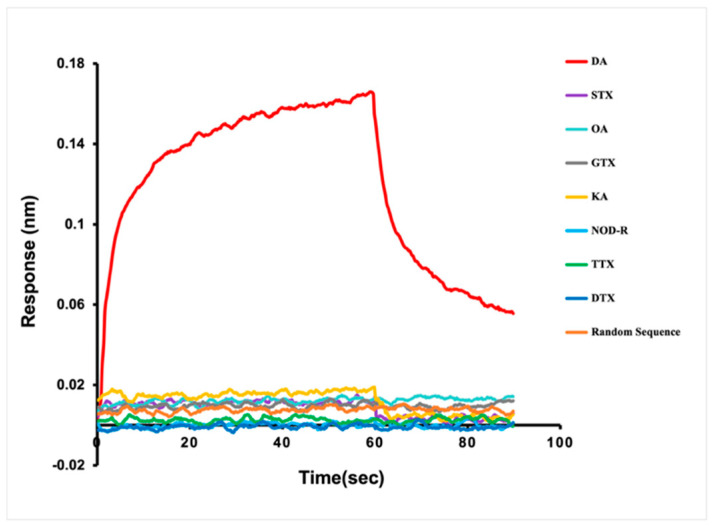
The affinity and specificity identification of C1-d for DA. The red, yellow, purple, cyan, gray, light blue, green, dark blue, and orange lines represent the binding curves of C1-d with DA, KA, saxitoxin (STX), okadaic acid (OA), gonyautoxin (GTX), nodularin-R (NOD-R), tetrodotoxin (TTX), dinophysistoxin (DTX), and random sequence, respectively. All the concentrations of toxins were 6.25 μM.

**Figure 5 bioengineering-09-00684-f005:**
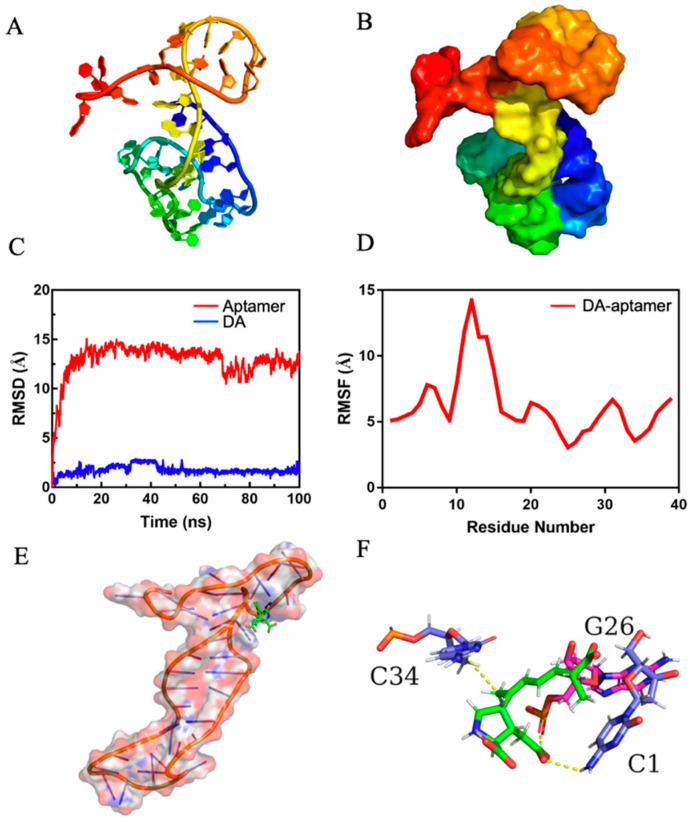
(**A**) Drawn view of C1-d 3D structure. (**B**) Surface view of C1-d 3D structure. (**C**) RMSD of DA–aptamer. (**D**) RMSF of DA–aptamer. (**E**) The binding model of DA–aptamer in the final stable structure. (**F**) The 3D binding mode of DA–aptamer. The aptamer was depicted as a cartoon and surface. The DA was colored green. The surrounding G was color in magenta and C was color in slate. The yellow dashed line shows hydrogen bonds.

**Figure 6 bioengineering-09-00684-f006:**
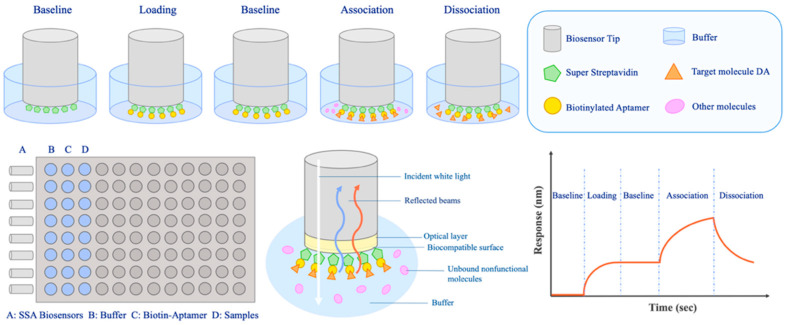
The working principle of biolayer interferometry.

**Figure 7 bioengineering-09-00684-f007:**
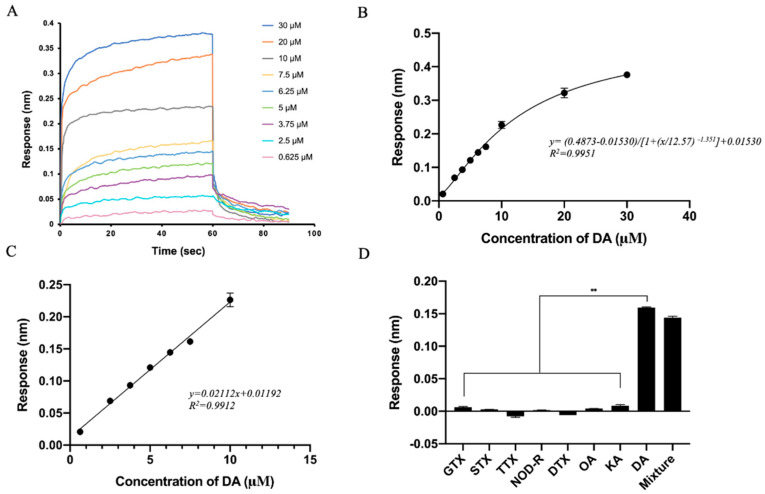
Evaluation of the BLI-based aptasensor. (**A**) Response value curve of the aptasensor at different concentrations of DA (0.625 to 30 μM). (**B**) Calibration curve of response values at different concentrations of DA (0.625 to 30 μM). (**C**) The linear range of the calibration curve of DA in the concentration of 0.625 to 10 μM. (**D**) The specificity of BLI-based aptasensor to various toxins (DA, GTX, STX, TTX, NOD-R, DTX, OA, KA each at 6.25 μM) and the mixture of the abovementioned toxins (the final concentration of each marine biotoxin was 6.25 μM). A random sequence was used as a control. The error bars represent standard deviations. ** *p* < 0.01 vs. DA (control).

**Table 1 bioengineering-09-00684-t001:** Affinity constants (*K*_D_) between candidate aptamers and DA.

No.	Aptamer Sequence (5′–3′)	*K*_D_ (M)
C1/htC1	ATTGGCACTCCACGCATAGGCCAACATGATGTTCCGTCATTTTGAGGTGTGTACACCGTGCCTATGCGTGCTACCGTGAA	1.69 × 10^−6^
C2	ATTGGCACTCCACGCATAGGGGATAACGGGTTGATGGTACTTCTATCTATCGCGTTGTGCCCTATGCGTGCTACCGTGAA	2.03 × 10^−5^
C12/htC2	ATTGGCACTCCACGCATAGGGACATCGAGAAGAATCCTGATACGACTTGGCTTTGCTGGCCCTATGCGTGCTACCGTGAA	5.21 × 10^−6^
C53	ATTGGCACTCCACGCATAGGGTAAAGTACGTATGTCATGCACATGCCGTATTCTCTTTGCCCTATGCGTGCTACCGTGAA	NB
C58/htC3	ATTGGCACTCCACGCATAGGCCAAGGTGTCAATCTTGAATAGCTGTGTAACGTCTATGTGCCTATGCGTGCTACCGTGAA	6.33 × 10^−6^
C66	ATTGGCACTCCACGCATAGGGGTGGTCTTCTGGGTAATCCCCGACTACCCTTATGCCGGCCCTATGCGTGCTACCGTGAA	2.27 × 10^−5^
C87	ATTGGCACTCCACGCATAGGGGAAGCTGCTCTCATCAATAAAAAATGAGACGGAGGTTACCCTATGCGTGCTACCGTGAA	5.04 × 10^−6^
C100/htC4	ATTGGCACTCCACGCATAGGGAATGGACCCGGTATAATTCCCTCAAGAGTG CCAATTTCACCTATGCGTGCTACCGTGAA	3.25 × 10^−6^
C116	ATTGGCACTCCACGCATAGGGATCTCATAACCAGTCTCTTTGACTGATGTTAGTAAGGTCCCTATGCGTGCTACCGTGAA	1.17 × 10^−5^
htC5	ATTGGCACTCCACGCATAGGGGAGGGCCGGTTGAACTGTAATGTGTAAACCACGCGCTACCCTATGCGTGCTACCGTGAA	NB
Random Sequence	ATTGGCACTCCACGCATAGGATTCCGGCTAATCGACTGACTGCCGGTACGATGCAGTCAGCCTATGCGTGCTACCGTGAA	NB

NB: No binding.

**Table 2 bioengineering-09-00684-t002:** The precision of aptasensors in different concentrations of DA (*n* = 3).

**DA (μM)**	2.5	3.75	5.0	6.25	7.5	10	20	30
**CV (%)**	4.63	1.97	1.50	1.12	1.30	4.71	4.47	1.39

**Table 3 bioengineering-09-00684-t003:** Detection of DA in real samples (*n* = 3).

Sample	DA (μM)	Detection Value (μM)	Recovery (%)	CV (%)
Seawater	5	5.48	109.6	3.28
	7.5	7.125	95	1.07
	10	10.89	108.9	3.98
Shellfish	5	4.95	98.9	2.3
	7.5	7.468	99.6	1.83
	10	11.03	110.3	1.45

**Table 4 bioengineering-09-00684-t004:** Current detection methods of DA.

Detection Method	Detection Object	LOD	Linear Range	Recovery Rate (%)	Reference
LC	mussel	20–30 μg/kg	over 10^4^	93	[13]
HPLC-UV	shellfish	25 ng/mL	50–5000 ng/mL	96–104	[15]
HPLC-UV	shellfish	500 μg/kg	—	72–92	[14]
PLE-LC–ESI-MS–MS	shellfish	200 μg/kg	50–5000 ng/mL	81–95	[63]
LC-MS	seawater and phytoplankton	0.03 ng/mL	0.05–400 ng/mL	95–104	[64]
RRLC-MS	seawater	0.02 ng/mL	—	92.1–110.6	[65]
cdELISA	shellfish	<25 μg/kg	—	73.8–92.8	[18]
ic-ELISA	shellfish	0.006 ng/mL	0.006–0.2 ng/mL	100.56 ± 2.8	[16]
Biosense Direct Competitive ELISA	shellfish	3.3 μg/kg	1.1–5.3 μg/kg	85.5–106.6	[17]
cELISA	blood	0.01 ng/mL	0.5–4.5 ng/mL	80	[19]
cITP-CZE	shellfish and algae	1.5 ng/mL	0–200 ng/mL	101 ± 3	[21]
CE-EIA	shellfish	0.02 ng/mL	0.1–50 ng/mL	89.6–105.8	[22]
Immuno-biosensor	shellfish	20,000 μg/kg	—	—	[60]
SPR biosensor	shellfish	3 μg/kg	4–60 μg/kg	—	[61]
MIP-SPR sensor	—	5 ng/mL	5–100 ng/mL	—	[62]

LC: Liquid chromatography; HPLC-UV: High-performance liquid chromatography–ultraviolet; PLE-LC–ESI-MS–MS: Pressurized liquid extraction–liquid chromatography–electrospray ionization mass spectrometry; LC-MS: Liquid chromatography–mass spectrometry; RRLC-MS: Rapid resolution liquid chromatography–mass spectrometry; cdELISA: Competitive direct enzyme-linked immunosorbent assay; ic-ELISA: Indirect competitive enzyme-linked immunosorbent assay; cELISA: Competitive enzyme-linked immunosorbent assay; cITP-CZE: Coupled capillary isotachophoresis–capillary zone electrophoresis; CE-EIA: Capillary electrophoresis-based enzyme immunoassay; MIP-SPR: Molecularly imprinted polymer–surface plasmon resonance.

## Data Availability

The data presented in this study are available in this article and Appendix A.

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
