# Peer review of "A Rapid and Sensitive Aptamer-Based Biosensor for Amnesic Shellfish Toxin Domoic Acid"

_bioengineering, 2022, doi:10.3390/bioengineering9110684_

Round 1

Reviewer 1 Report

In general, this study design was well-delineated and reported.

The corrections required are mainly allocated to the introduction, as the authors are not well familiarized with ASP monitoring and its public health aspects:

Line 37- the term ‘pollution’ is not the preferred designation for ‘harmful microalgae blooms’ (HABs). The word pollution generally implies that the contaminants have an anthropogenic source, while HABs are natural phenomena (although in some cases enhanced by anthropogenic nutrients, like in China)

Line 58- ‘when’ seems to be missing here: «it was found that the content of DA in shellfish reached 40 mg/kg can cause poisoning»

Line 59: «the maximum limit of human tolerance is 20 mg/kg» This is not correct. This is a regulatory limit, aimed at avoiding human exposures above a certain level. It applies to shellfish only. Human exposure is another metric and is expressed in body intake, it is 30 µg DA/kg b.w. (body weight)

Line 64: «At present, the detection methods of DA mainly include mouse bioassay (MBA),» etc. This is not correct, as the bioassay was judged unfit for monitoring purposes decades ago (in the 1990’s). The routine method is based on HPLC-UV detection around the world. See comment for line 75. Also, there is another major shortcoming in using the mouse bioassay acid extracts (even for HPLC): the toxin is not stable in very low pH. Never mentioned in the introduction so far.

Line 75: The authors failed to cite the most relevant article for the analysis of domoic acid: Rapid Extraction and Cleanup for Liquid Chromatographic Determination of Domoic Acid in Unsalted Seafood; Journal of AOAC INTERNATIONAL, Volume 78, Issue 2, 1 March 1995. All monitoring programs around the world are mainly based on this method, which requires HPLC-ultraviolet detection (much cheaper than LC-MS equipment). References 16 and 17 are HPLC methods which were based on hplc-fluorimetry, that were probably only used by those authors themselves when developing the methods – nobody else followed those methods. Other methods rather than the HPLC-UV I refer above, are more complex or require equipment not ready available (or more expensive) as the straight measurement by UV detection.

Line 96: «there are still some defects in current DA detection techniques.» The word ‘defects’ is not appropriate. ‘Shortcomings’ is more often used. There is no need to attack other commonly used and quite accessible screening methods, such as HPLC-UV. No effort was made to criticize other methods from the point of view of green chemistry: for example, HPLC uses a lot of acetonitrile. Not so green. What about the aptamer method?

The « molecular recognition» methods also require a lot of equipment. Is it cheaper than HPLC-UV equipment? The equipment used was not mentioned in the M&M section adequately. Please mention more clearly the contents provided in the supplementary material at the begin of section 2.

Line 115: « apply them to the detection and treatment of DA» What is meant here by treatment? Treating human poisoning? Or preventing it in the first place by a monitoring approach?

Line 322: the panoply of toxin acronyms was not explained so far. What were the sources of these chemicals? No adequate mention in the text to the supplementary section.

Line 482: »the low cost of aptamers and reusability of aptasensors» How is it possible to reuse? By washing the plate?

Reviewer 2 Report

The authors selected aptamers towards domoic acid (DA).  The aptamer that was found the most times among candidates randomly selected from the 12th round of selection, and that also was the most enriched when the pools were examined by high throughput sequencing had the best KD of the candidates that were characterized and was chosen for integration into a biolayer interferometry (BLI) based assay.  First the sequence was truncated and one truncation let to an improved KD.  This version was demonstrated in the BLI format.

I found parts of this paper difficult to follow because of poor English.  Other parts could be improved with more complete descriptions. 

1. Why was the C1-s chosen for the molecular docking and molecular dynamics simulations?  Wouldn’t it have been more relevant to do these with the C1-d?

2. Is there a way to test variants of the aptamer to test the model experimentally and to confirm the regions that are predicted to have interactions with the DA?  For example, could C1 be changed to a different base to determine if that results in a lowered affinity?  Please explain why the C1-a is able to bind DA (KD of 6.9uM) when it is missing 2 of the 3 positions speculated to form hydrogen bonds with the DA, and all but one of the 7 positions suggested to contribute most to the complex binding free energy.    

3. There are parts of the truncation sequences that were not explained.  Why were two Cs added to the front of C1-d that are not found in that part of the aptamer sequence?  Why is the end of C1-c  AGTG while the analogous sequence in C1-s is AGGTG? 

4. What is the error on the KD measurement?  Were replicate measurements performed?

5. What do the colors mean in panels A and B of figure 5?

6.  Is there an advantage to use an aptamer for small molecule detection using a BLI sensor over an antibody or an scFv?

7.  Although the authors describe the C1-d as high affinity, a KD of 109 nM is not very high compared to typical antibody affinities.  Is the poorer affinity an advantage in this format where the fast off rate enables reuse of the sensor probe? 

8.  It would be helpful to show lower concentrations of DA down to ~50 nM DA (or below) to demonstrate that the sensor was experimentally able to detect values around the theoretical LOQ.

9. When discussing detection of seawater (LOD, LOQ, dynamic range) the authors change from molar to ug/mL.  Please keep concentrations units consistent.  Also, it could be helpful to add the standard curves in seawater and shellfish samples to the supplemental information file.

10.  Panel C of Figure 7 needs a label on the X-axis.

11. The paragraph from line 453 to 472 discusses downsides of other methods for detecting DA.  Many of the “issues” with the other methods are not completely fair.  There is a way to show the advantages of the current BLI assay without totally misrepresenting the other methods.  For example the authors point out that HPLC requires expensive equipment.  Although this may be true, a BLI instrument capable of analyzing 96 samples simultaneously and able to detect interactions with small molecules is also costly!  The authors state that ELISA is complex and “needs to be operated by professional personnel”.  We routinely have student interns successfully performing ELISAs in our laboratory.  These are teenagers who are not professional personnel.  Maybe just focus on the advantages of the current method and leave out most (or all) of the rest.

12.  Likewise in the next paragraph (starting on line 473, the authors state that in their method there is “no tedious sample preparation.”.  However, the methods section of this paper describes methanol water extraction of the DA in shellfish samples followed by evaporation, resuspension, and filtering. 

13.  Table 4 would be most helpful to have consistent units for comparison of the methods.  The currently developed BLI assay should also be included in the table for easy reference.

14.  What model BLI instrument was used in this work? 

Round 2

Reviewer 1 Report

I am satisfied with the improvments made to the first version. The authors folowed the reviewer sugestions.

Reviewer 2 Report

The authors have mostly addressed my concerns, but some of the information they included in their response would be helpful to include in the supplemental information.   Although I had the revised version of the manuscript, I did not have access to the revised version of the supplemental information.  (I realized I had only the original version of the supplemental information file because the sequence of C1-s3 had not been corrected in table S3 or figure S5, and there was no figure S9). 

1. Please consider adding a table to the supplemental information that presents the KD data provided in the response to my comment #4 for C1-s, C1-sa, C1-sb, C1-Sc, and C1-sd.

2. Please consider adding the figure provided in the answer to my comment 2 to the supplemental information file.

3.  Please consider adding the data table provided in response to my comment #8, along with the values for a sample with no added DA, (and potentially values for higher concentrations if the authors choose) to the supplemental material.  Including the response for a sample with no DA is imperative in order to show meaningful response at low concentrations.
